# Shear Bond Strength of a Direct Resin Composite to CAD-CAM Composite Blocks: Relative Contribution of Micromechanical and Chemical Block Surface Treatment

**DOI:** 10.3390/ma15145018

**Published:** 2022-07-19

**Authors:** Vincent Fouquet, François Lachard, Sarah Abdel-Gawad, Elisabeth Dursun, Jean-Pierre Attal, Philippe François

**Affiliations:** 1Innovative Dental Materials and Interfaces Research Unit (URB2i, UR4462), Faculty of Health, Paris Cité University, 1 Rue Maurice Arnoux, 92120 Montrouge, France; francois.lachard@u-paris.fr (F.L.); sarah.abdel-gawad@u-paris.fr (S.A.-G.); elisabeth.dursun@u-paris.fr (E.D.); jean-pierre.attal@u-paris.fr (J.-P.A.); philippe.francois@u-paris.fr (P.F.); 2Department of Prosthetic Dentistry, Louis Mourier Hospital, 178 Rue des Renouillers, 92700 Colombes, France; 3Department of Pediatric Dentistry, Henri Mondor Hospital, 1 Rue Gustave Eiffel, 94000 Créteil, France; 4Department of Dental Materials, Charles Foix Hospital, 7 Avenue de la République, 94200 Ivry-sur-Seine, France; 5Department of Dental Materials, Bretonneau Hospital, 23 Rue Joseph de Maistre, 75018 Paris, France

**Keywords:** bond strength, CAD-CAM blocks, polymer infiltrated ceramic network, dispersed filler composite, universal primer

## Abstract

This study aims to compare the shear bond strength (SBS) of a direct resin composite to CAD-CAM resin composite blocks treated with different surface treatments: micromechanical, chemical or a combination of both. Eight CAD-CAM resin composite blocks, namely Brilliant Crios, Cerasmart 270, Vita Enamic, Grandio block, Katana Avencia, Lava Ultimate, Tetric CAD and Shofu Block HC were chosen. The micromechanical surface treatment protocols tested were hydrofluoric acid, polyacrylic acid or sandblasting, and the chemical one was a universal primer. These treated CAD-CAM blocks were tested to determine the SBS of a light-curing composite resin Z100 bonded to their surface. Two-way ANOVA followed by Tukey’s post hoc test was used to investigate the difference in SBS. Failures were analyzed by Fisher’s exact test. Bonding interfaces were examined by scanning electron microscopy. The micromechanical surface treatments give the highest SBS values: sandblasting appears to be the most efficient procedure for dispersed filler composite blocks, while hydrofluoric acid etching is preferable for polymer-infiltrated ceramic network (PICN) blocks. The use of universal primer does not improve SBS values on dispersed filler composite blocks. For PICN blocks, the use of universal primer significantly increases SBS values when combined with hydrofluoric acid etching.

## 1. Introduction

CAD-CAM resin composite blocks make it possible to perform partial or full coverage restorations. They promote chairside restorations that lead to reduced work time, which helps in the prevention of SARS-CoV-2 contaminations [1]. These CAD-CAM resin composite blocks overcome the drawbacks of direct composites, such as insufficient curing or shrinkage stress during light-curing [2], especially on direct posterior restorations [3]. Their biocompatibility with the cells of the oral tissues is better than that of direct composites due to their polymerization under high pressure and high temperature during their manufacturing processes [4,5]. Even if ceramic blocks have better mechanical properties [6], dispersed filler resin composite blocks present a better fatigue resistance [7], higher machinability [8], easier finishing properties, and are less abrasive for antagonist teeth [9]. In addition, the emergence of polymer-infiltrated ceramic networks (PICNs) or hybrid materials with specific mechanical properties, such as an elastic modulus intermediate between CAD-CAM dispersed filler composites and glass-ceramics [10], expands the therapeutic arsenal in restorative dentistry by offering an alternative to dispersed charge composites.

However, a few years ago, the manufacturer of Lava Ultimate (3M ESPE), a resin composite block with dispersed fillers, withdrew the crown indication due to a reportedly high debonding rate [11]. When bonding on tooth structures, there are two interfaces: dental tissues/resin luting cement and resin luting cement/CAD-CAM resin restoration [12]. To strengthen the latter, several surface treatments of CAD-CAM resin restorations have been proposed: creation of micromechanical retentions, such as sandblasting and hydrofluoric acid etching, which are the most well-known procedures [13,14,15], and chemical adhesion through silanes or functional monomers [14,15]. Polyacrylic acid (PA) is known to react to the aluminosilicate network by breaking the glass network [16]. This PA feature could be effective on the dispersed glass filler found in composite resins. No consensus has been reached regarding the optimal surface treatment. 

Moreover, to our knowledge, there are still few studies investigating the shear bond strength of a light-curing viscous composite resin bonded to CAD-CAM resinous materials, even if it could lead to high bonding values by reducing the elastic modulus mismatch between the dental tissue and the composite resin used as a bonding material [17,18,19]. Few studies have also compared the relative importance of micromechanical and chemical adhesion on the shear bond strength of a composite resin to a CAD-CAM resin block [20,21].

The aim of this in vitro study was to assess the relative contribution of micromechanical and chemical surface treatments to the shear bond strength of a viscous direct composite resin to CAD-CAM resin blocks.

## 2. Materials and Experimental Procedures

### 2.1. Samples Preparation

Seven commonly used CAD-CAM dispersed filler composite resin blocks and 1 PICN were tested. All information is given in Table 1.

For each material, a slice of 8 mm by 12 mm with 2-mm-thick of the material were made in CAD-CAM blocks using a low-speed diamond saw (Isomet, Buehler, Coventry, UK) with constant water application, oriented perpendicular to the major axis of the block. These slices were then embedded in self-curing acrylic resins (Plexcil-Escil, Chassieu, France) in plastic cylinders (diameter: 25 mm, depth: 15 mm), exposing one of its larger surfaces, then polished with sandpaper (800 grits) using a polishing machine (Planopol 3, Struers, Kobenhavn, Denmark) under water cooling. Lastly, the samples were cleaned by ultrasonication.

### 2.2. Surface Treatments

For each material, samples were randomly assigned to eight subgroups (8 × *n* = 12): no further surface treatment, sandblasting, hydrofluoric acid (HF), polyacrylic acid (PA), universal primer, sandblasting + universal primer, HF + Universal primer and PA + Universal Primer, belonging to four main groups: control group, micromechanical retention, chemical retention, and a combination of both.

Manufacturers, batch numbers and composition of the products used are also noted in Table 1. The surface treatments and the main groups to which they belong are presented in Table 2.

For each subgroup, ten samples were used to perform shear bond strength (SBS) tests and two were used for scanning electron microscopy examination (SEM) analysis.

### 2.3. Shear Bond Strength (SBS) Tests and Failure Mode Determination

After adequate surface treatment, a cylindrical Teflon mold was placed on each sample to build a 4 mm-high cylinder of composite resin in two increments (diameter = 3 mm), with a flat base of 7 mm^2^. After the material was light-cured with a polywave curing light with a minimum output of 950 mW/cm^2^ (Valo Grand Cordless, Ultradent Products, South Jordan, UT, USA), the mold was removed and the excess material, if present, was gently eliminated from around the base of the material cylinder with a scalpel.

The SBS was determined using a universal testing machine (Lloyd Instruments, Fareham, UK). The shear force was applied at the CAD-CAM resin block/composite resin interface, with a chisel-shaped blade parallel to the block surface. A cross-head speed of 0.5 mm/min was chosen.

The debonded specimens were observed under a binocular microscope (BZH10 Olympus, Hamburg, Germany) at ×30 magnification and the failure modes were classified into the following five types:−CF-B: cohesive in the block material if more than 75% of fracture regards it;−AF: adhesive if more than 75% of the block surface is intact and free of resin composite;−MF: mixed if the intact surface free of the block is between 25 and 75%;−CF-RC: cohesive in resin composite if more than 75% of fracture regards the resin composite.

### 2.4. Scanning Electron Microscopy Examination (SEM)

After the surface treatment, two samples of each group tested were air-dried and metalized with gold (Sputter-Coater, Bio-Rad, Marnes-la-Coquette, France) for microscopic examination at 1000× magnification (JSM-6400, JEOL LTD, Tokyo, Japan).

### 2.5. Statistical Analysis

Normal distribution was confirmed by the Shapiro–Wilk test, and the equality of variances was assessed using the Levene test before the tests were performed. SBS data were expressed as mean values and standard deviations. A two-way ANOVA followed by Tukey’s post hoc test was used to investigate the difference in SBS between the different surface treatments. Failure modes were analyzed by Fisher’s exact test for single comparisons between groups and in pairwise analysis.

In all tests, the significance level was set at *p* < 0.05. Statistical calculations were performed using R Software (v3.6.1; R Foundation for Statistical Computing, Vienna, Austria).

## 3. Results

### 3.1. SBS Values and Failure Mode

The SBS values for the experimental groups: control group, micromechanical retention, chemical retention and a combination of both, are summarized in Table 3.

#### 3.1.1. SBS for Micromechanical Retention Group

Two-way ANOVA revealed that SBS was significantly influenced by the micromechanical surface treatment. The highest SBS value was obtained for Vita Enamic, the only PICN material after hydrofluoric acid etching (44.94 MPa), whereas the highest SBS value for dispersed filler CAD-CAM composite resins was obtained for Katana Avencia after sandblasting (43.75 MPa).

Sandblasting has been shown to be the most effective surface treatment to increase the SBS value of CAD-CAM dispersed filler blocks, with an average gain of 38.20%, whereas hydrofluoric acid surface treatment leads to an average gain of 18.18%.

For the Vita Enamic Block (PICN), sandblasting surface preparation leads to a gain of 24.16% of the shear bonding strength compared to the control group, but hydrofluoric acid etching was the best surface treatment, with a gain of 57.40% compared to the control group.

#### 3.1.2. SBS for Chemical Retention Group

The two-way ANOVA revealed no significant differences between the different groups.

#### 3.1.3. SBS for Combination of Micromechanical and Chemical Retention Group

The two-way ANOVA revealed that SBS was significantly influenced by micromechanical treatment followed by chemical surface treatment.

The highest SBS value among all the sub-groups was obtained for Vita Enamic after hydrofluoric acid etching and universal primer application (46.44 MPa), whereas the highest SBS value for dispersed fillers CAD-CAM composite resin was obtained for Katana Avencia after sandblasting and universal primer application (44.09 MPa). The SBS value of Vita Enamic after hydrofluoric acid etching followed by universal primer surface treatment (micromechanical + chemical) was statistically significantly higher than hydrofluoric acid etching only (micromechanical), whereas no statistically significant benefit could be demonstrated with the use of a universal primer after the micromechanical surface treatments.

The lowest SBS value among all the sub-groups was obtained for Shofu Block HC without any pre-treatment (26.60 MPa), whereas the lowest SBS value for Vita Enamic was obtained without any pre-treatment (28.55 MPa).

#### 3.1.4. Failure Mode

Failure mode were cohesive in the CAD-CAM block in every group. No significant differences were shown between the various groups.

### 3.2. SEM Analysis

SEM images at 1000× magnification for all surface treatments are shown in Figure 1, Figure 2, Figure 3, Figure 4, Figure 5, Figure 6, Figure 7 and Figure 8 for each block. For all the materials tested, striations were visible in the control group, corresponding to the 800 grits polishing performed.

The universal primer, sandblasting and hydrofluoric acid etching led to changes in appearance for all tested materials. For CAD-CAM dispersed filler composite blocks, the modification was created by sandblasting, whereas, for PICN, the highest modification was created by hydrofluoric acid etching.

## 4. Discussion

This study only investigated the immediate bonding of a composite resin to the surface of pretreated CAM-CAM blocks, and no conclusion can be drawn regarding the possible contribution of long-term treatment with a universal primer. Few studies are currently available on this subject [22].

### 4.1. Bond Strength to CAD-CAM Composite Block

Surprisingly, in the absence of pre-treatment, when the block was polished to 800 grits, bonding values between 30.2 MPa (Grandio Block) and 26.6 MPa (Shofu Block HC) were obtained for the different blocks tested. This point was reinforced by the fact that all the observed failure modes were cohesive. This observation could be due to the fact that a 800 grits polishing roughness corresponds to the finish obtained with a red-ring diamond bur [23]. This significant roughness is suggested by the SEM images for different groups, as previously described, and is able to provide correct micromechanical bonding with direct resin composites and can be improved with higher surface roughness [24]. Moreover, the conversion rate of chain-growth polymerization in a CAD-CAM resin composite blocks is much higher than that of a direct resin and easily exceeds 95% [25]. This suggests a low possibility of copolymerization with the remaining free methacrylic monomers of the block with the bonding resin composite. This observation of the correct bonding strength values with an 800 grits polishing procedure is directly linked to the micromechanical component of adhesion. 

### 4.2. Failure Modes on CAD-CAM Blocks Bonding

Cohesive failures were observed in all the tested CAD-CAM material groups. While many studies analyze cohesive fractures in the tested material as the mark of adhesion values exceeding the cohesive strength of the material, cohesive failure is explained by other parameters like mechanics of the test or mechanical parameters of materials [17,18,26]. Larger bonding areas, like in macro-shear bond strength performed in this study, have been shown to have a higher probability of presenting a critical flaw that could lead to a catastrophic failure and thus cohesive failure [27]. Moreover, finite element studies showed that macro-retention elements could improve adhesion with the consequence of shifting from adhesive failure profiles (without macro-retention) to cohesive or mixed failure profiles [28]. The same mechanism can be considered for the micro-retention created by the surface treatments on CAD-CAM dispersed fillers or PICN blocks. This could imply a different stress distribution in the material and explain the fact that significant differences for SBS values were obtained between certain surface treatments for a given block despite that only cohesive fractures were observed.

### 4.3. Effect of Micromechanical Surface Treatment

Sandblasting significantly increased the SBS values for the dispersed filler composites and PICN. The aim of air abrasion is to increase the surface roughness of the material by creating random irregularities in the matrix, thereby increasing the surface energy [2,29]; this also improves the wettability of the resin composite, which leads to better anchoring by micro-retention [30]. Sandblasting a CAD-CAM resin also has other advantages, such as cleaning the bonding surface after saliva contamination due to a try-in simulation by providing a fresh and clean surface [31]. Numerous studies have attempted to find an optimal sandblasting protocol by varying the pressure or duration of treatment. The optimal treatment would be sandblasting between 1 and 3 bar for 5 to 60 s, depending on the study and the type of composite [32,33]. However, it has been shown that sandblasting may damage the surface of composite CAM-CAM blocks if the procedures (pressure, duration, type of abrasive particle) are not optimized [34]. Severe cracks may appear inside the resin matrix and at the interface between the resin matrix and filler particles. Moreover, the cracks may propagate, causing premature catastrophic failures [30]. In view of our study, it can be considered a key factor in improving the bond strength of a composite resin to CAD-CAM dispersed filler composite blocks.

However, its influence on Vita Enamic (PICN) is weaker. One explanation could be the higher surface hardness of Vita Enamic [35] than that of dispersed filler composites, leading to a lower abrasivity on PICNs in a given time. Etching with hydrofluoric acid for 60 s, according to the manufacturer’s instructions and other publications [36,37] is its most effective surface treatment by dissolving the feldspathic (glass-containing) ceramic skeleton [38]. The polymer network of a composite is not altered by hydrofluoric acid and has no deleterious effect on the covalent bonds of methacrylate polymeric macromolecules. Thus, hydrofluoric etching allows the creation of increased roughness by the dissolution of silica (SiO_2_) in the feldspathic skeleton. The typical honeycomb SEM images visible after hydrofluoric acid application on Vita Enamic (Figure 3) allow excellent micromechanical anchorage [39]. The percentage of hydrofluoric acid and the application time also influenced the adhesion on PICN. The 10% hydrofluoric acid creates a higher roughness than 5% hydrofluoric acid. In addition, a 60 s application would give better results than a 20 s application [22,40,41]. For PICNs, hydrofluoric acid is therefore shown to be a key factor in obtaining high adhesion values.

To explain the lower efficiency of hydrofluoric acid treatment on CAD-CAM composites with dispersed fillers, we can argue that the vast majority of fillers are not composed of pure SiO_2_. Their mineral compositions contain crystalline or polycrystalline ceramics that are less sensitive or insensitive to hydrofluoric acid [42,43]. For the composite blocks containing only SiO_2_-based fillers, the volume percentage of the inorganic phase was lower than that of the PICN, which led to a lower micromechanical component after etching.

Polyacrylic acid is used in the surface treatment of dental tissues to optimize the adhesion of GICs or self-adhesive resin cement [44]. Therefore, we hypothesized that polyacrylic acid could not only improve the adhesion of composite blocks by gently attacking the fillers included in the composite blocks, but also clean the prosthetic surface from all forms of contamination. Our results did not reveal significant differences with the control group for all the materials. It is advisable not to use this surface treatment on composite blocks.

### 4.4. Effect of Chemical Surface Treatment

Since silane-only products are being phased out in favor of universal primers, and universal adhesives have been widely shown to be less effective in treating prosthetic intrados than universal primers [45,46,47], only the effect of a universal primer was investigated in this study.

Some authors have reported better results after using different prosthetic primers such as silane, universal primer (containing silane and 10-MDP), or the use of a universal adhesive; however, their use remains debated in the literature [21,48]. Expected improvements for universal primers or universal adhesives could be linked to the adhesion of the silane to the SiO_2_ exposed fillers or 10-MDP to polycrystalline exposed fillers.

Increasing the resin wettability by silane application has also been proposed to enhance adhesion to CAD-CAM resin blocks [2]. In our study, no improvement in SBS values of any CAD-CAM resin block was observed after universal primer use, as in other studies [49,50]. The application of this universal primer alone, or the following micromechanical component, is relatively time-consuming for dental practice. Hence, the overall working time is more than a minute. This protocol has to be performed carefully by application for 60 s and a perfect drying of the universal primer. During the activation of the silane primer, a polycondensation reaction begins, leading to the creation of water molecules [51]. If the final drying sequence is improperly performed, the risk of contamination with water during the luting procedure to resin CAD-CAM blocks is high. This risk is truly operator-dependent and could be avoided by canceling this step during the bonding sequence because of the lack of statistical difference in our study when no universal primer was used on CAD-CAM composite resin blocks.

### 4.5. Effect of Combination of Micromechanical and Chemical Surface Treatment

Some authors have described the positive effects of primers containing silane after air abrasion [52,53]. Others consider silane beyond sandblasting as the key factor for resin composite bonding [34,54]. We were unable to demonstrate it. Indeed, we obtained comparable SBS values with micromechanical surface treatment alone and with micromechanical combined with chemical treatments.

In the case of PICNs, the current opinion is that the use of a primer containing silane after treatment with hydrofluoric acid significantly increases adhesion during bonding [36] which has also been verified in this study. The universal primer facilitates the infiltration of the adhesive into the crevices of the material [55]. However, some authors have obtained contradictory results in which the benefits of silane have not been demonstrated [22].

## 5. Conclusions

The micromechanical component of block surface treatment appears to be the most important factor in improving the immediate SBS of a resin composite. To create the required surface roughness, sandblasting is the most efficient procedure for dispersed filler composite blocks, while hydrofluoric acid etching is preferable for PICN resin blocks. Universal primers containing 10-MDP and silane did not improve the immediate SBS values of CAD-CAM resin blocks with dispersed filler. The combination of micromechanical and chemical surface treatment significantly improves SBS values for PICN blocks compared to the micromechanical component alone.

However, further in vitro and in vivo studies are necessary to assess the impact of aging on these procedures.

## Figures and Tables

**Figure 1 materials-15-05018-f001:**
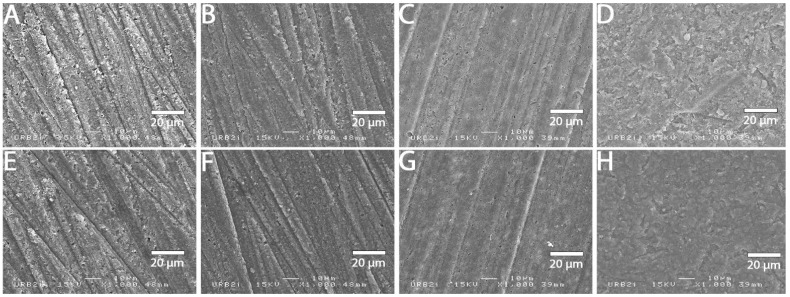
SEM micrograph for Brillant Crios surface (×1000 magnification) with the different surface treatments: (**A**) control; (**B**) polyacrylic acid; (**C**) hydrofluoric acid; (**D**) sandblasting; (**E**) 800 grit polishing + universal primer; (**F**) polyacrylic acid + universal primer; (**G**) hydrofluoric acid + universal primer; (**H**) sandblasting + universal primer. Sandblasting surface treatment (**D**,**H**) shows a more irregular pattern surface than the other ones.

**Figure 2 materials-15-05018-f002:**
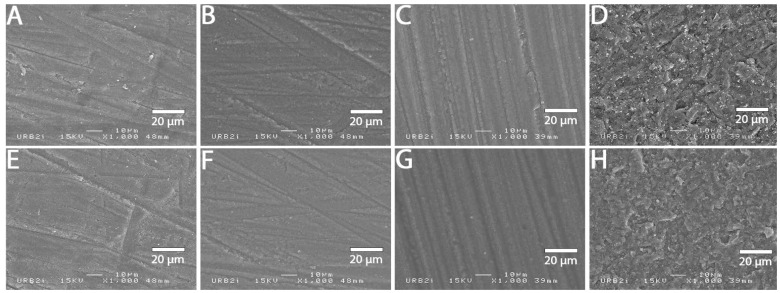
SEM micrograph for Cerasmart 270 surface (×1000 magnification) with the different surface treatments: (**A**) control; (**B**) polyacrylic acid; (**C**) hydrofluoric acid; (**D**) sandblasting; (**E**) 800 grit polishing + universal primer; (**F**) polyacrylic acid + universal primer; (**G**) hydrofluoric acid + universal primer; (**H**) sandblasting + universal primer. Sandblasting surface treatment (**D**,**H**) shows a more irregular pattern surface than the others ones.

**Figure 3 materials-15-05018-f003:**
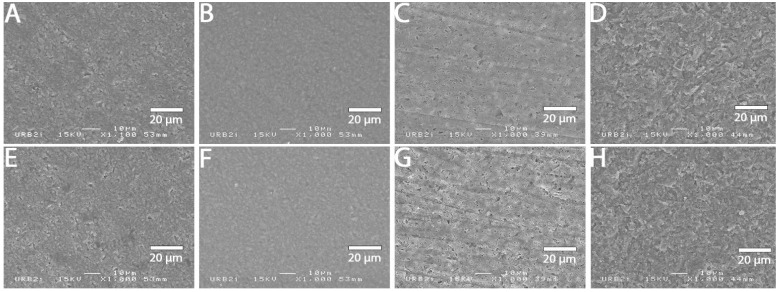
SEM micrograph for Grandio block surface (×1000 magnification) with the different surface treatments: (**A**) control; (**B**) polyacrylic acid; (**C**) hydrofluoric acid; (**D**) sandblasting; (**E**) 800 grit polishing + universal primer; (**F**) polyacrylic acid + universal primer; (**G**) hydrofluoric acid + universal primer; (**H**) sandblasting + universal primer. Sandblasting surface treatment (**D**,**H**) shows a deeper and more irregular pattern surface than the other ones.

**Figure 4 materials-15-05018-f004:**
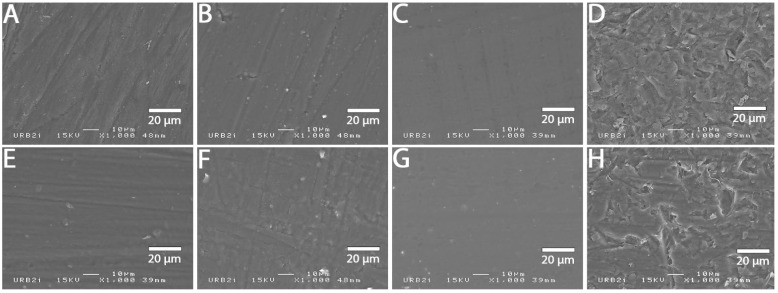
SEM micrograph for Katana Avencia surface (×1000 magnification) with the different surface treatments: (**A**) control; (**B**) polyacrylic acid; (**C**) hydrofluoric acid; (**D**) sandblasting; (**E**) 800 grit polishing + universal primer; (**F**) polyacrylic acid + universal primer; (**G**) hydrofluoric acid + universal primer; (**H**) sandblasting + universal primer. Sandblasting surface treatment (**D**,**H**) shows a more irregular pattern surface than the other ones.

**Figure 5 materials-15-05018-f005:**
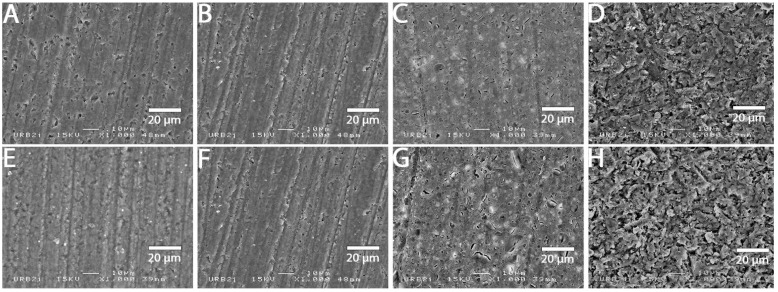
SEM micrograph for Lava Ultimate surface (×1000 magnification) with the different surface treatments: (**A**) control; (**B**) polyacrylic acid; (**C**) hydrofluoric acid; (**D**) sandblasting; (**E**) 800 grit polishing + universal primer; (**F**) polyacrylic acid + universal primer; (**G**) hydrofluoric acid + universal primer; (**H**) sandblasting + universal primer. Sandblasting surface treatment (**D**,**H**) shows a more irregular pattern surface than the other ones.

**Figure 6 materials-15-05018-f006:**
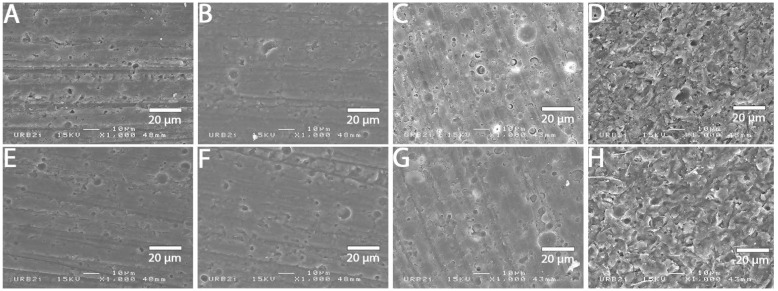
SEM micrograph for Shofu Block HC surface (×1000 magnification) with the different surface treatments: (**A**) control; (**B**) polyacrylic acid; (**C**) hydrofluoric acid; (**D**) sandblasting; (**E**) 800 grit polishing + universal primer; (**F**) polyacrylic acid + universal primer; (**G**) hydrofluoric acid + universal primer; (**H**) sandblasting + universal primer. Sandblasting surface treatment (**D**,**H**) shows a more irregular pattern surface than the other ones.

**Figure 7 materials-15-05018-f007:**
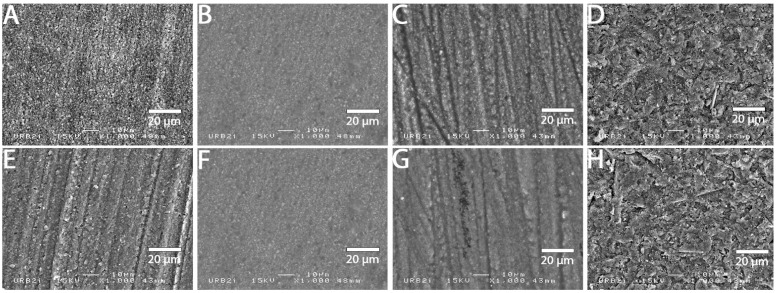
SEM micrograph for Tetric CAD surface (×1000 magnification) with the different surface treatments: (**A**) control; (**B**) polyacrylic acid; (**C**) hydrofluoric acid; (**D**) sandblasting; (**E**) 800 grit polishing + universal primer; (**F**) polyacrylic acid + universal primer; (**G**) hydrofluoric acid + universal primer; (**H**) sandblasting + universal primer. Sandblasting surface treatment (**D**,**H**) shows a more irregular pattern surface than the other ones.

**Figure 8 materials-15-05018-f008:**
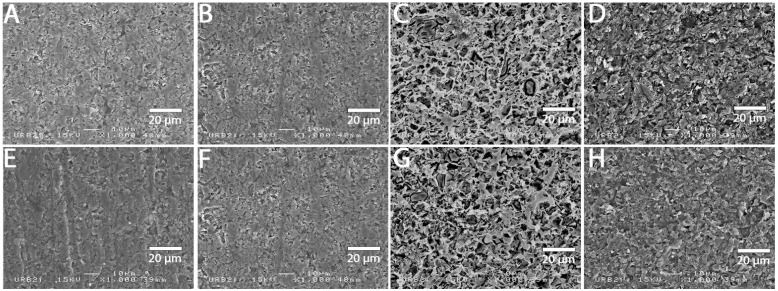
SEM micrograph for Vita Enamic surface (×1000 magnification) with the different surface treatments: (**A**) control; (**B**) polyacrylic acid; (**C**) hydrofluoric acid; (**D**) sandblasting; (**E**) 800 grit polishing + universal primer; (**F**) polyacrylic acid + universal primer; (**G**) hydrofluoric acid + universal primer; (**H**) sandblasting + universal primer. Hydrofluoric acid surface treatment (**C**,**G**) shows a more irregular pattern surface than the other ones.

**Table 1 materials-15-05018-t001:** Abbreviations, manufacturers, batch numbers and composition of the materials tested.

Materials		Abbreviation	Manufacturer	Batch Number	Composition
**CAD-CAM blocks**	Brilliant Crios (dispersed filler resin block)	**BC**	Coltene-Whaledent, Altstatten, Switzerland	I28626	Bis-GMA, BIS-EMA, TEGDMA, 71 wt% barium glass, and silica particles
	Cerasmart 270 (dispersed filler resin block)	**CS**	GC Corporation, Tokyo, Japan	2002276	UDMA, Bis-MEPP, DMA, 71 wt% silica, barium glass
	Grandio Block (dispersed filler resin block)	**GB**	Voco, Cuxhaven, Germany	1810664	Cross-linked dimethacrylate, 86 wt% inorganic filler.
	Katana Avencia (dispersed filler resin block)	**KA**	Kuraray-Noritake, Niigata, Japan	000740	UDMA, TEGDMA, 62 wt% Al_2_O_3_ and SiO_2_
	Lava Ultimate (dispersed filler resin block)	**LU**	3M ESPE, St. Paul, MN, USA	N721285	Bis-GMA, Bis-EMA, UDMA, TEGDMA, 80 wt% silica and zirconia nanoparticles, zirconia and silica nanoclusters
	Tetric CAD (dispersed filler resin block)	**TC**	Ivoclar-Vivadent, AG, Schaan, Liechtenstein	X55553	Cross-linked dimethacrylate, 80 wt% nanoparticles
	Shofu Block HC (dispersed filler resin block)	**SH**	Shofu, Kyoto, Japan	0819919	UDMA, Bis-EMA, TEGDMA, 80 wt% SiO_2_ and ZrO_2_ particles, and aggregated ZrO_2_/SiO_2_-nanoclusters
	Vita Enamic (PICN)	**VE**	Vita-Zahnfrabrik, Bad Säckingen, Germany	78100	UDMA, TEGDMA, 86 wt% sintered network (SiO_2_, Al_2_O_3_, Na_2_O, K_2_O, B_2_O_3_, ZrO_2_, CaO)
**Direct light-curing resin composite**	Z100	**Z100**	3M ESPE, St Paul, MN, USA	NA59173	Bis-GMA, TEGDMA, 2-benzotriazolyl-4-methyphenol 2, 84,5 wt% Zirconia/Silica
**Universal Primer**	Monobond Plus	**MP**	Ivoclar-Vivadent, AG, Schaan, Liechtenstein	Y39580	Ethanol, 3-trimethoxysilylpropyl methacrylate, 10-MDP (MDP), sulfide methacrylate.
**Hydrofluoric Acid**	Porcelain Etch	**HF**	Ultradent Products, South Jordan, UT, USA	BKYL4	Buffered 9.0% hydrofluoric acid.
**Polyacrylic Acid**	Dentin Conditioner	**PA**	GC Corporation, Tokyo, Japan	1902121	Distilled water, polyacrylic acid.

**Table 2 materials-15-05018-t002:** Surface treatment performed on CAD-CAM blocks.

Group Tested	Specific Protocol
**Control**	Control group
**SB** **(micro mechanical retention group)**	CAD-CAM block surfaces were sandblasted with 50 µm aluminum oxide at 2 bar pressure for 20 s at a 90° angle and a distance of 10 mm. Then, samples were cleaned by ultrasonication and dried with oil-free air.
**HF** **(micro mechanical retention group)**	CAD-CAM block surfaces were etched with Hydrofluoric acid for 60 s and rinsed thoroughly with water. Finally, samples were dried with a strong stream of oil-free air.
**PA** **(micro mechanical retention group)**	CAD-CAM block surfaces were cleaned with polyacrylic acid for 60 s and rinsed thoroughly with water. Finally, samples were dried with a strong stream of oil-free air.
**MP** **(chemical retention group)**	Universal primer Monobond plus was applied on CAD-CAM block surfaces for 60 s and dried with a strong stream of oil-free air.
**SB + MP** **(micro mechanical and chemical retention group)**	CAD-CAM block surfaces were sandblasted with 50 µm aluminum oxide at 2 bar pressure for 20 s at a 90° angle and a distance of 10 mm. Then, the sample was cleaned by ultrasonication and dried with oil-free air. Finally, Monobond plus was applied on sample surfaces for 60 s and dried with a strong stream of oil-free air.
**HF + MP** **(micro mechanical and chemical retention group)**	CAD-CAM block surfaces were etched with Hydrofluoric acid for 60 s and rinsed thoroughly with water. Then, sample were cleaned by ultrasonication and dried with a strong stream of oil-free air. Finally, Monobond plus was applied on sample surfaces for 60 s and dried with a strong stream of oil-free air.
**PA + MP** **(micro mechanical and chemical retention group)**	CAD-CAM block surface was cleaned with polyacrylic acid for 60 s and rinsed thoroughly with water. Then, the sample was cleaned by ultrasonication and dried with a strong stream of oil-free air. Finally, Monobond plus was applied on sample surfaces for 60 s and dried with a strong stream of oil-free air.

**Table 3 materials-15-05018-t003:** Mean and standard deviations of shear bond strength (SBS) after micromechanical, chemical and combination of micromechanical and chemical surface treatment.

Material Tested	Subgroup	SBS in MPa (±SD)
Brillant Crios	Control	**30.57** (±2.28) **^J,K,L,M,N,O^**
	Sandblasting	**38.58** (±1.31) **^B,C,D,E,F,G,H,I^**
	HF	**33.21** (±2.47) **^H,I,J,K,L,M,N^**
	PA	**30.57** (±3.23) **^J,K,L,M,N,O^**
	Universal Primer	**30.67** (±4.21) **^J,K,L,M,N,O^**
	Sandblasting + MP	**39.49** (±2.43) **^B,C,D,E,F,G,H^**
	HF + MP	**33.74** (±2.80) **^H,I,J,K,L,M,N^**
	PA + MP	**30.14** (±2.31) **^J,K,L,M,N,O^**
Cerasmart 270	Control	**30.50** (±3.08) **^J,K,L,M,N,O^**
	Sandblasting	**41.59** (±2.04) **^A,B,C,D,E,F^**
	HF	**35.52** (±2.35) **^F,G,H,I,J,K,L^**
	PA	**30.03** (±3.11) **^J,K,L,M,N,O^**
	Universal Primer	**31.12** (±3.44) **^J,K,L,M,N,O^**
	Sandblasting + MP	**42.01** (±2.16) **^A,B,C,D,E^**
	HF + MP	**35.75** (±2.35) **^D,E,F,G,H,I,J,K,L^**
	PA + MP	**30.39** (±2.88) **^J,K,L,M,N,O^**
Grandio Block	Control	**30.65** (±3.07) **^J,K,L,M,N,O^**
	Sandblasting	**40.89** (±3.08) **^A,B,C,D,E,F,G^**
	HF	**32.82** (±2.50) **^I,J,K,L,M,N,O^**
	PA	**30.96** (±2.63) **^J,K,L,M,N,O^**
	Universal Primer	**30.65** (±3.07) **^J,K,L,M,N,O^**
	Sandblasting + MP	**42.27** (±2.46) **^A,B,C^**
	HF + MP	**33.45** (±2.76) **^H,I,J,K,L,M,N^**
	PA + MP	**31.17** (±2.55) **^J,K,L,M,N,O^**
Katana Avencia	Control	**29.52** (±3.52) **^L,M,N,O^**
	Sandblasting	**43.75** (±2.93) **^A,B^**
	HF	**35.66** (±1.95) **^D,E,F,G,H,I,J,K,L^**
	PA	**30.26** (±2.34) **^J,K,L,M,N,O^**
	Universal Primer	**30.45** (±3.76) **^J,K,L,M,N,O^**
	Sandblasting + MP	**44.09** (±2.81) **^A,B^**
	HF + MP	**36.47** (±1.89) **^C,D,E,F,G,H,I,J^**
	PA + MP	**30.51** (±2.34) **^J,K,L,M,N,O^**
Lava Ultimate	Control	**29.80** (±3.57) **^L,M,N,O^**
	Sandblasting	**39.07** (±2.24) **^B,C,D,E,F,G,H,I^**
	HF	**32.83** (±2.11) **^I,J,K,L,M,N,O^**
	PA	**30.30** (±3.36) **^J,K,L,M,N,O^**
	Universal Primer	**31.04** (±2.64) **^J,K,L,M,N,O^**
	Sandblasting + MP	**41.01** (±2.48) **^A,B,C,D,E,F,G^**
	HF + MP	**33.09** (±2.56) **^H,I,J,K,L,M,N^**
	PA + MP	**31.20** (±1.82) **^J,K,L,M,N,O^**
Shofu Block HC	Control	**26.60** (±2.69) **^O^**
	Sandblasting	**41.13** (±2.03) **^A,B,C,D,E,F,G^**
	HA	**34.96** (±2.45) **^G,H,I,J,K,L,M^**
	PA	**28.13** (±2.76) **^N,O^**
	Universal Primer	**28.40** (±3.27) **^N,O^**
	Sandblasting + MP	**42.11** (±2.09) **^A,B,C,D^**
	HA + MP	**35.64** (±2.59) **^E,F,G,H,I,J,K,L^**
	PA + MP	**29.81** (±1.94) **^L,M,N,O^**
Tetric CAD	Control	**29.90** (±3.89) **^K,L,M,N,O^**
	Sandblasting	**41.80** (±2.27) **^A,B,C,D,E,F^**
	HF	**31.97** (±2.70) **^J,K,L,M,N,O^**
	PA	**30.69** (±2.33) **^J,K,L,M,N,O^**
	Universal Primer	**30.41** (±3.73) **^J,K,L,M,N,O^**
	Sandblasting + MP	**42.54** (±2.47) **^A,B,C^**
	HF + MP	**33.24** (±2.11) **^H,I,J,K,L,M,N^**
	PA + MP	**30.90** (±2.38) **^J,K,L,M,N,O^**
Vita Enamic	Control	**28.55** (±2.02) **^M,N,O^**
	Sandblasting	**35.45** (±2.27) **^F,G,H,I,J,K,L^**
	HA	**44.94** (±2.29) **^A,B^**
	PA	**30.65** (±2.84) **^J,K,L,M,N,O^**
	Universal Primer	**29.95** (±3.80) **^K,L,M,N,O^**
	Sandblasting + MP	**36.29** (±2.23) **^C,D,E,F,G,H,I,J,K^**
	HA + MP	**46.44** (±2.73) **^A^**
	PA + MP	**30.07** (±2.36) **^J,K,L,M,N,O^**

Values with the same upper case superscript letter are not significantly different at *p* < 0.05 (Tukey’s test).

## Data Availability

Data available on request.

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
