# Peer review of "Shear Bond Strength of a Direct Resin Composite to CAD-CAM Composite Blocks: Relative Contribution of Micromechanical and Chemical Block Surface Treatment"

_materials, 2022, doi:10.3390/ma15145018_

Round 1

Reviewer 1 Report

Dear Authors,

The manuscript reflects the robust nature of the study and depicts the outcome of your investigation in an appropriate manner. However, I have some specific comments to be addressed. 

Reviewer 2 Report

This study measured the shear bond strength (SBS) of a direct resin composite to CAD-CAM resin composite blocks treated by different surface treatments. For study of bond strengths, roughness measurement, aging or thermal cycling are important experiments that should be included. The followings are some specific comments.  

1.      Introduction: The authors did not clearly describe the reason of measuring the bond strength of a direct resin composite to CAD-CAM resin composite blocks. As they mentioned there are two interfaces: den[1]tal tissues/resin luting cement and resin luting cement/CAD-CAM resin restoration, why the authors did not measure the bond strength of resin luting cement to CAD-CAM resin restoration?

2.      The description of 3.2. SEM Analysis was too short. The important findings of SEM observation were not found in text or figure legends. In addition, there should be space between adjacent SEM pictures to let the readers clearly observe.

3.      Sandblasting was performed at a 45° angle and a distance of 10 mm. How the angle and distance was controlled? The roughness might not be evenly distributed due to the 45° angle of performance.

Reviewer 3 Report

A major criticism is the SBS values are not coherent with the fact that all fractures occurred in the composite blocks. For cohesive fractures, the measured values did not represent the bond strength, but the inherent strength of the block material.

This needs to be ruled out. One possible explanation is that the fracture mode evaluation was incorrect. It is difficult to differentiate between composite remnant and composite block at low magnification (30x), and especially if there shades are similar or near the same.

Another less likely explanation is that the surface treatment affect not only the surface but also deeper in the composite blocks.

For SBS testing, it is critical that the knife edge is place in the area where the two materials is united. Placing the knife edge away from this area, the forces applied is no longer sheer, but bending. The reviewer suspects this could also have influenced the results.

Spesific comments

Abstract:

#1 Spell out PICN («for PICN resin blocks”)

Introduction, 4th line:

#2 What is “ they “ referring to ([2], they present a). Please rephrase.

#3 Please give the rationale for investigating the bond strength between composite blocks and a direct composite resin material. More relevant had been resin based cements.

2.1. Samples preparation:

#4 Delete first paragraph. All information was given in Table 1.

2.2. Surface Treatments

#5 It was stated that one group had “no surface treatment”. This must be incorrect, as in the paragraph above it was stated “polished with sandpaper (800 grits)”

Please correct to “no further surface treatment”

Table 2:

#6 Delete second column as it gives no extra information from what was found in the other two columns.

#7 Table 3 is over-complex:

#8 Use the abbreviations for table 1 consistently in 2nd column

#9 Delete 3rd column (information was found in table 2)

#10 Move data for Vita Enamic to end of Table 3. Place the other materials in alphabetical order.

#11 4th column: Place results of statistical calculations (upper case letters) after mean±SD

#12 If the data in Table 4 was correct (please se comment#), delete the table and replace with one sentence.

#13 Figures 1-8:

Place white lines between photos. Add size bar.

Below figure 8:

#14 It is stated “The universal primer, sandblasting, and hydrofluoric acid etching increases the surface roughness for all tested materials”.

How was this estimated? Any measurements? What can be seen from the Figures, is changes in appearance, but not that this represents changes in surface roughness. Please add measurements.

Round 2

Reviewer 2 Report

The manuscript can be accepted in the revised form.

Author Response

The authors thank the reviewer for his proofreading. No modifications have been done on the manuscript.

Reviewer 3 Report

The added paragraph does not solve the major critisim, that all spesimens fractured cohesively in the substrate but the different surface treatments resulted in different SBS values.

Author Response

The authors thank the reviewer for his proofreading. 

The following paragraph has been modified in the text for more clarity :

"Larger bonding areas, like in macro-shear bond strength performed in this study have been shown to have a higher probability of presenting a critical flaw that could lead to a catastrophic failure and thus cohesive failure [23]. Moreover, finite element studies showed that macro-retention elements could improve adhesion with the consequence of shifting from adhesive failure profiles (without macro-retention) to cohesive or mixed failure profiles [24]. The same mechanism can be considered for the micro-retention created by the surface treatments on CAD-CAM dispersed fillers or PICN blocks. This could imply a different stress distribution in the material and explain the fact that significant differences for SBS values were obtained between certain surface treatments for a given block despite that only cohesive fractures were observed."

This has been fixed in blue in the manuscript.